# UHT Cow’s Milk Supplementation Affects Cell Niches and Functions of the Gut–Brain Axis in BALB/c Mice

**DOI:** 10.3390/biomedicines12112448

**Published:** 2024-10-25

**Authors:** Felipe S. Lemos, Caio A. Prins, Ana M. B. Martinez, Raul Carpi-Santos, Arthur S. Neumann, Nathalia Meireles-da-Costa, Roberto Luisetto, Valeria de Mello-Coelho, Felipe L. Oliveira

**Affiliations:** 1Immunopharmacology, Oswaldo Cruz Institute, Oswaldo Cruz Foundation (Fiocruz), Rio de Janeiro 21040-360, Brazil; simoeslemos.f@gmail.com; 2Department of Pathology, Faculty of Medicine, Federal University of Rio de Janeiro, Rio de Janeiro 21941-913, Brazil; caioprins.neuro@gmail.com (C.A.P.); anamartinez@hucff.ufrj.br (A.M.B.M.); 3Institute of Biomedical Sciences, Federal University of Rio de Janeiro, Rio de Janeiro 21941-901, Brazil; raulcarpi@gmail.com (R.C.-S.); arthurneumann19@gmail.com (A.S.N.); coelhova@histo.ufrj.br (V.d.M.-C.); 4Molecular Carcinogenesis Program, Brazilian National Cancer Institute, Rio de Janeiro 20230-130, Brazil; nathalia.meireles@inca.gov.br; 5Department of Surgery, Oncology and Gastroenterology, University of Padova, 35124 Padova, Italy; roberto.luisetto@unipd.it

**Keywords:** gut–liver axis, cow’s milk, intestinal inflammation, systemic inflammation, neuroinflammation, behavioral patterns

## Abstract

Background/Objectives: Cow’s milk is a bioactive cocktail with essential nutritional factors that is widely consumed during early childhood development. However, it has been associated with allergic responses and immune cell activation. Here, we investigate whether cow’s milk consumption regulates gut–brain axis functions and affects patterns of behaviors in BALB/c mice, previously described by present low sociability, significant stereotypes, and restricted interest features. The major objectives consist of to investigate cow’s milk supplementation as possible triggers interfering with cellular niches of the gut–brain axis and behavioral patterns. Methods: Male BALB/c at 6 weeks were randomly divided into two groups, one supplemented with cow’s milk processed at ultra-high temperature (UHT) and another group receiving water (controls) three times per day (200 μL per dose) for one week. Results: Milk consumption disturbed histological compartments of the small intestine, including niches of KI67^+^-proliferating cells and CD138^+^ Ig-secreting plasma cells. In the liver, milk intake was associated with pro-inflammatory responses, oxidative stress, and atypical glycogen distribution. Milk-supplemented mice showed significant increase in granulocytes (CD11b^+^SSC^high^ cells) and CD4^+^ T cells in the blood. These mice also had neuroinflammatory signals, including an enhanced number of cortical Iba-1^+^ microglial cells in the brain and significant cerebellar expression of nitric oxide synthase 2 by Purkinje cells. These phenotypes and tissue disorders in milk-supplemented mice were associated with atypical behaviors, including low sociability, high restricted interest, and severe stereotypies. Moreover, synaptic niches were also disturbed after milk consumption, and Shank-3^+^ and Drebrin^+^ post-synaptic cells were significantly reduced in the brain of these mice. Conclusions: Together, these data suggest that milk consumption interfered with the gut–brain axis in BALB/c mice and increased atypical behaviors, at least in part, linked to synapse dysfunctions, neuroinflammation, and oxidative stress regulation.

## 1. Introduction

Cow’s milk is a popular beverage and a staple in many diets worldwide. It is a bioactive cocktail containing microRNAs, hormones, carbohydrates, lipids, and vesicles [1,2]. In essence, milk is digested and broken down into its component nutrients, which are then absorbed through the gut mucosa into the bloodstream. The process involves various mechanisms to ensure effective nutrient uptake and may also influence gut health and immune function through its bioactive compounds [3]. In the stomach, proteins such as casein are broken down into smaller peptides and amino acids [4]. In the small intestine, enzymes like lactase break down lactose (milk sugar) into glucose and galactose [5].

There are conflicting effects attributed to cow’s milk consumption. Although several benefits to health have been described after milk intake [6], possible correlations between milk consumption, neuroinflammation and development of neurodegenerative diseases, neuropsychiatric disorders, or atypical behaviors have been reported [7,8].

Indeed, milk can influence the gut–brain axis through its nutrient content, impacting gut microbiota, immune system interactions, and potential effects on stress and mood [9]. The gut–brain axis is increasingly recognized as a key factor in various neurological and psychological disorders [10]. In this context, the gut–brain axis may influence behavioral patterns through mechanisms like gut microbiota imbalance, inflammation, neurotransmitter production, and dietary factors.

BALB/c mice can be considered an interesting model to study the triggers that potentialize atypical behaviors. These mice naturally have autistic-like signals, including low sociability, self-aggressive behaviors, and stereotypes when compared with neurotypical C57BL/6 mice [11]. In the current work, BALB/c mice were challenged with cow’s milk and the gut–liver–brain axis was partially analyzed. Here, the data suggested that cow’s milk is a possible trigger for atypical behaviors in BALB/c, such as autistic as signals, which include stereotypes, sociability, and restrict interest.

## 2. Materials and Methods

### 2.1. Animals and Milk Supplementation

Six-week-old male BALB/c mice were obtained from the colony bred at the Federal University of Rio de Janeiro, Brazil. The experimental protocols were approved by the local Animal Ethics Committee of the Federal University of Rio de Janeiro (protocol number 071/19), Brazil, in accordance with the guidelines provided by the Brazilian College of Animal Experimentation in rooms with controlled temperature and light/dark 12-h cycles. Mice were randomly separated in two groups maintained with water and food ad libitum: (I) mice supplemented with 200 µL three times/day (600 µL/day) of UHT cow’s milk by oral gavage during 7 consecutive days and (II) controls supplemented with water at similar conditions. This study was carried out in compliance with the ARRIVE guidelines. Each experimental group was composed of 10 mice.

### 2.2. Behavioral Assays

Prior to behavior and motor assays, all animals were acclimated to the testing rooms and to the testing platforms for 5 days. Each day they rested during 1 h inside every new room and subsequently adapted inside the testing platforms: 4 min inside open-field; 10 min inside the three-chambered test platform without the novel colored cubic toy and familiar or unfamiliar mouse; 40 min alone inside the acrylic cage for stereotyped motor behavior filming; and, for the ladder rung walk, adaptation they needed just to walk freely over the metal rods for 5 min or complete 3 circuits. These tests were performed in sound-isolated rooms and on alternative days, to avoid emotional distress and autonomic-freezing responses. Each mouse was recorded for 600 s (10 min) after the period of adaptation to the analysis platforms (5 min). In the three-chamber test, sociability was measured by the time spent in the chamber containing another animal (familiar or unfamiliar), or even an object previously presented to the animals. Regarding stereotypes, we used a stopwatch to add the time spent by each animal performing a repetitive movement on the open-field platform. The graphs were created, and the evaluated time was described on the Y axis in seconds. Data were plotted as the average of the tested populations.

### 2.3. Social Interaction, Social Play, and Social Retraction

Each group of mice at a time were placed inside an open arena 90 cm × 45 cm × 7 cm box filled with nesting material, and they were recorded at 5 frames/second for 15 min from a webcam situated 60 cm above the box. Attitudes of social interaction and social play such as anal sniffing, nose sniffing, pouncing, and pinning were recorded for each animal, and the seconds spent on these behaviors were quantified over the total filmed time. Social retraction, meaning the condition of non-interaction plus isolation from a group of a vigil and a not-moving mouse, was also counted and the percentage of time over the total filmed time was likewise quantified.

### 2.4. Stereotyped Motor Behavior

Each mouse was individually recorded for 15 min and 5 frames/second from a webcam positioned 60 cm in front of a translucent acrylic cage with 10 cm × 10 cm × 15 cm dimensions to monitor stereotyped motor attitudes such as paw flapping, persistent self-grooming, circling, and jumping. We quantified the time of each persistent stereotyped motor behavior and summed all time spent on these over the total filmed time.

### 2.5. Three-Chamber Test

To assess sociability and interest in social or spatial novelty, we performed this experiment using one testing mouse (control or milk-treated) that would be presented at the same time to (1) a cage containing a mouse already recognized by the testing mouse and native from the same cage and from the same group, and (2) a cage containing a colored cubic toy not previously presented. To check the sociability and social–spatial interest, we counted the time spent by the testing mouse near the cage with the already-known same-cage mice or with the never-presented colored cubic toy (understood here as 5.0 cm from the nose tip of the testing mouse and pointing toward one of the cage-edge directions) or the time spent by the testing mouse touching, sniffing, and exploring one of these cages. This experiment was recorded for 10 min by a webcam, 5 frames/second, and positioned 1.5 m above the cage. We quantified the exploration time over the total recorded time.

### 2.6. Basso Mouse Scale, BMS

The test was performed inside a 90 cm circular open-field acrylic platform (EP 154B Insight Equipamentos Ltd.a-EPP, BR). Each animal was evaluated for four minutes by two observers in a double-blinded manner, and were graded from 0 to 9, where 0 represents total hind-limb locomotor impairment with paraplegia and 9 is the normal hind-limb locomotor function. Their performance inside the open-field arena was also recorded with a webcam positioned 1.50 m above the platform and 5 frames/second for 2 min. Using ImageJ 1.x, we quantified the percentage of the explored perimeter from the start of outside movement to the open-field center. For such, three points were selected: (1) the center of the open field where the animal was initially located (O); (2) the first point where the animal touched and started to explore the perimeter of the open field (I); (3) the farthest angular point from the initial perimeter exploration point (F). The angle formed by these three points (*FÔI*) is the angular distance from the open-field perimeter. We quantified the percentage of this exploration angular distance over the total angular perimeter (360°) and used the same software to track the mouse’s route over the platform.

### 2.7. Ladder Rung Walk Test

This test was carried out inside a horizontally straight 50 cm ladder platform with 3 mm diameter metal rods spaced each other from a 5 mm gap. This catwalk-like platform has 90 spaces for metal rod insertion and using the online software RANDOM.ORG (Dublin, Ireland): True Random Number Service, version Interger Generator, available at https://www.random.org (accessed on 11 December 2019), we randomly selected 60 entrances for the metal rod insertion, creating a random pattern and gaps between the metal rods. This test is used to evaluate the fine motor control of mice limbs. These same movements were analyzed to identify possible disturbances in the sensibility of paws, including protopathic sensibility. Each mouse was filmed by a 5 frames/second webcam during three u-turn routes over the ladder platform. We quantified the percentage of correct steps over the metal rods and divided this number by the number of total steps (total steps number = correct steps over the metal rods + slips between the metal rods gaps).

### 2.8. Histological Analysis and Immunohistochemistry

Gut, liver, and brains were collected, immediately fixed in 4% buffered formalin for 72 h at room temperature and segmented into cortex and cerebellum fragments. For hematoxylin and eosin (H & E) staining, samples were washed three times in xylene and rehydrated in reduced concentrations of ethanol (100 to 70%). Then, samples were stained sequentially with Harris hematoxylin and eosin. For mounting, slides were dehydrated and covered with Entellan (Sigma-Aldrich, St Louis, MO, USA) and glass coverslips. For periodic acid–Schiff (PAS) staining, samples were deparaffinized, rehydrated, and washed for 5 min with periodic acid–Schiff, and then the slides were dehydrated and covered with Entellan and glass coverslips.

For immunohistochemistry, the sections were heated in Trilogy^TM^ (Sigma-Aldrich, St Louis, MO, USA). Peroxidase and non-specific antibody interactions were inhibited by 3% H_2_O_2_ and BSA 8% diluted in 0.002% PBS-Tween 20; primary antibodies: anti-SHANK3 (Sigma-Aldrich, USA), anti-NOS2, and anti-IBA-1 (BD Bioscience, Billerica, MA, USA). After secondary antibody incubation, samples were treated with streptavidin-peroxidase (Sigma-Aldrich, St Louis, MO, USA), diaminobenzidine (DAB) used as substrate, and samples counterstained with Harris’ hematoxylin. The percentage of positive cells was defined as the number of marked cells in 500 cells per sample. Morphological analysis was performed using high-power microscopy (Zeiss-Axioplan, Zeiss, Jena, Germany) and images were acquired by Evolution MP 5.0 RTV-Color camera (Media Cybernetics, Surrey, BC, Canada).

### 2.9. Immunofluorescence

Paraffin sections were deparaffinized with 2 xylene washes for 15 min each. After deparaffinization, the sections were submitted to a gradient of ethanol solution (100% to 50%) of 2 min each. Then, samples were incubated with PBS 1x/0.05% between 20 and 30 min. Afterward, the sections were washed with PBS 1x/0.05% Tween 20, incubated in citrate buffer at 98 °C for 40 min, and treated with blocking buffer (3% bovine serum albumin, 5% normal goat serum (Sigma-Aldrich, USA) diluted in PBS 1X/0.05% Tween 20). After blocking, the samples were incubated with the primary antibodies overnight, washed 3 times with PBS 1X/0.05% Tween 20, and incubated with secondary antibodies. Nuclei were counterstained with DAPI (Sigma-Aldrich, USA). Slices were mounted with DAKO Mounting Media and imaged on a confocal microscope (Leica TCS SPE, Wetzlar, Germany). Primary antibodies: anti-synaptophysin and anti-drebrin (Milipore, Burlington, MA, USA). Secondary antibodies: anti-Mouse 594 for synaptophysin and Anti-Rb 488 for drebrin (Molecular Probes, Oregon, USA). After capturing 6–8 images per group of cortical external granular layer, the green and red channels were aligned, and the co-localized puncta were analyzed using the Puncta Analyzer plug-in in NIH ImageJ. Experiments were performed in duplicate.

### 2.10. Statistical Analysis

The statistical tests were accomplished using *t*-test, and the significance threshold was fixed for α = 0.05. Therefore, *p* values ≤ 0.05 were considered statistical differences. Each experiment was performed using 6 mice per group in three independent assays.

## 3. Results

### 3.1. Cow’s Milk Administration Was Associated with Autistic-like Behaviors in BALB/c Mice

BALB/c mice have been used as an animal model for identifying genes and neurobiological pathways involved in autism-related phenotypes, including dietary products that interfere with autism-related disorders in the gut–brain axis [11,12]. To investigate the effects of UHT milk consumption on the gut–brain axis functions, we evaluated behavioral patterns in these mice comparing the effects of milk supplementation with control mice receiving water. Then, these mice were submitted to three-chamber tests. Milk-supplemented mice showed significant reduction in social interactions with a familiar mouse and unfamiliar mouse (Figure 1A). These supplemented mice spent more time performing repetitive stereotyped behaviors than the respective control mice (Figure 1B). Furthermore, milk-supplemented mice were associated with low interest to explore new objects (Figure 1C) and, consequently, these mice showed more restricted interest to known objects (Figure 1D). These data indicated that milk consumption was associated with significant autistic-like behaviors in BALB/c mice.

The Global Motility Test performed in the circular arena revealed that control mice moved through practically every free space in the arena, but mice that received cow’s milk showed significant alteration in the patterns of movement and exploration (Figure 1E). Intriguingly, these milk-supplemented mice moved with apparent anxiety-related behavior, preferring the border of the arena. They moved in repeated paths and when they returned to the starting point, they explored the neighboring area a little more (Figure 1E).

The Global Motility Test revealed that control mice explored approximately 90–95% of the perimeter while milk-supplemented mice explored only 50% of perimeter in the same arena (Figure 1F). Importantly, milk consumption did not interfere with displacement velocity during the exploration of the arena (Figure 1G). Other relevant data revealed that total wandered distance was similar between both groups of mice (Figure 1H), suggesting that differences between explored areas by controls and milk-treated mice would be a consequence of fear, insecurity, anxiety, or any other similar central block after milk intake.

Motor coordination and sensibility tests were also carried out to assess possible grosser damages. Thus, mice were submitted to Motor Function Assessment and Fine Motor Skill Assessment. Both groups presented similar scores concerning motor functions (Figure 1I) and fine motor skills (Figure 1J). The sensibility in paws was also analyzed in mice that received cow’s milk. In comparison with controls, cow’s milk intake did not interfere with protopathic sensibility (Figure 1K). These data suggested that motor coordination and sensibility in paws were not affected by cow’s milk consumption by the BALB/c mice.

### 3.2. Oral Administration of Milk Affected the Distribution of Blood Leukocytes

To correlate milk supplementation and behavioral patterns, some compartments were individually investigated in both mice groups, including blood, gut, liver, and brain. The peripheral blood samples from both experimental groups were evaluated by flow cytometry. The phenotype of the cells was defined by the type of membrane protein present in each cell type, for example, CD11b for myeloid cells (granulocytes and monocytes), B220 for B lymphocytes, CD4 for helper T lymphocytes, and CD8 for cytotoxic T lymphocytes. The granularity pattern was essential to detect cell subtypes with similar phenotypes. In the blood of control animals, approximately 12% of leukocytes were detected as granulocytes (CD11b^high^SSC^high^). In mice supplemented with milk, this percentage rose to approximately 45% of leukocytes with the CD11b^high^SSC^high^ phenotype, indicating a significant increase in granulocytes in the blood of animals supplemented with milk (Figure 2A,B). CD11b^low^SSC^low^ monocytes did not show significant differences in the percentage obtained from control and milk-treated animals (Figure 2A,C).

In lymphoid cell populations, the consumption of cow’s milk modified B lymphocytes and CD4^+^ T lymphocytes. In control mice, approximately 28% of leukocytes were B lymphocytes, characterized by the B220^+^SSC^low^ phenotype (Figure 2D,E). On the other hand, in mice supplemented with milk, this percentage decreased to approximately 12% of leukocytes, indicating that B220^+^SSC^low^ B lymphocytes were significantly affected by milk consumption (Figure 2D,E).

CD4 T lymphocytes represented approximately 10% of circulating leukocytes in control animals. This percentage rose to 23% of leukocytes in the blood of mice supplemented with cow’s milk (Figure 2F,G). However, the percentage of CD8^+^ T lymphocytes was not affected by milk consumption (Figure 2F,H). These data indicate that milk affected the distribution of granulocytes, B lymphocytes, and CD4^+^ T lymphocytes, indicating a potential to deregulate the immune system in BALB/c mice.

### 3.3. Milk Consumption Induced Gut Inflammation in Mice

The second compartment analyzed was the small intestine of mice supplemented with milk or water. In the controls, typical histological layers were identified as mucosa, submucosa, and external smooth muscle tissue (Figure 3A). The small intestine of milk-supplemented mice also showed these histological structures, but there were significant signals of inflammatory reactions in the mucosa, such as enlargement of villi with vacuolated enterocytes and hyperplasia of the crypts (Figure 3B). Intraepithelial lymphocytes were quantified in the gut mucosa of the controls and milk-supplemented mice to monitor signals of inflammatory reaction, including cow’s milk intolerance and allergy [13,14]. The small intestine of milk-treated mice has a higher percentage of intraepithelial lymphocytes than control mice (Figure 3C). This significant increase in intraepithelial lymphocytes was associated with other inflammatory niches, such as KI67^+^ cells and Ig-secreting plasma cells in the gut.

The hyperplasia of the crypts was subsequently investigated by the KI67 proliferative marker. In control mice, KI67^+^ cells were preferentially observed in the crypts, although rare cells were also detected in the villus (Figure 3C). Milk-supplemented mice showed a dispersive organization of these KI67^+^-proliferating cells, including robust follicular nodules throughout the mucosa (Figure 3D). These data reinforced the histopathological findings concerning cryptal hyperplasia in the small intestine after milk consumption. Another indicator of inflammatory signals in the gut was the organization of plasma cells expressing CD138 (Syndecan-1). In control mice, CD138^+^ plasma cells were frequently found in the base or intermediate area of the villus (Figure 3E). In milk-supplemented mice, the distribution of CD138^+^ plasma cells was significantly perturbed throughout the villus (Figure 3F). At least in part, the higher cellularity inside these villi after milk consumption seems associated with the reorganization of CD138^+^ plasma cells.

### 3.4. Milk Consumption Induced Hepatic Oxidative Stress in Mice

The abnormal organization of the gut mucosa led us to investigate the liver, focusing on the gut–liver axis. In the controls, the liver presented normal histological parameters. Hepatocytes were characterized by enlarged cells with round nucleus (sometimes two nuclei), eosinophilic cytoplasm and polygonal format making up approximately 80% of the liver mass. Moreover, Kupffer cells were also frequently observed within the hepatic sinusoids (Figure 4A). Milk consumption induced significant histological changes in the hepatocytes, including chromatin compaction, nuclear decentralization, and cytoplasmic pallor with a vacuolated appearance (Figure 4B).

Although histological signals of inflammation were undetectable in the liver after milk supplementation, samples of both control and supplemented mice were submitted for analysis of enzyme nitric oxide synthase 2 (NOS2). In control mice, the hepatocytes showed low levels of NOS2 expression by their cells, being predominantly perinuclear and little diffused throughout the hepatic lobe (Figure 4C). On the other hand, mice supplemented with milk had a high intensity of staining for NOS2 in the cytoplasm of hepatocytes, with broad diffusion of NOS2^+^ cells throughout the liver parenchyma (Figure 4D).

To investigate the origin of the vacuolated appearance of hepatocytes after milk consumption, liver samples were stained with the PAS (periodic acid–reactive Schiff) method. The images indicate that glycogen is abnormally accumulated in the liver of the milk-supplemented mice when compared to the control animals (Figure 4E,F). Together, these data indicated that milk induced metabolic disturbances in the liver of BALB/c mice.

### 3.5. Milk Intake Dysregulated the Expression of Genes Linked to Gut–Liver Axis Functions

These previous findings led us to propose that the gut–liver axis could be altered in the organizational context. Thus, we investigated the expression of genes involved in early inflammatory events and/or tissue repair. In the liver of BALB/c mice supplemented with cow’s milk, IL-1β, CCL2, and CXCL1 mRNA levels were significantly increased when compared to control mice (Figure 5A, Figure 5B and Figure 5C, respectively). These data indicate that gene expression of pro-inflammatory cytokines and chemokines are enhanced after the milk consumption. In contrast, the expression of the CX3CL1 mRNA was similar between both experimental conditions (Figure 5D). CYP1A2 mRNA levels were also increased in the liver of milk-supplemented mice (Figure 5E). In accordance, this gene codifies one of the most abundant hepatic cytochrome P450 enzymes, directly associated with metabolism of several foods and drugs [15]. In contrast, CYP1A1 mRNA did not modify in the liver after milk consumption (Figure 5F). The expression of aryl hydrocarbon receptor (AhR) mRNA was also unmodified in milk-supplemented mice (Figure 5F). This receptor functions as a key regulatory factor that modulates environmental, dietary, or microbial signals [16]. However, the enhanced expression of glucocorticoid receptor alpha (GRa) mRNA indicates that the liver is receiving inflammatory signals derived from the gut (Figure 5G). This receptor mediates the tissue-specific metabolic and immune responses under stressful conditions, including LPS challenge [17]. Together, these findings suggest that milk intake can induce pro-inflammatory signals in the liver of BALB/c mice, perhaps affecting the gut–liver axis.

### 3.6. Gut–Liver Axis Disturbances Were Linked to Neuroinflammatory Signals After Milk Intake

Given that milk intake disturbed the gut–liver axis and induced an imbalanced leukocyte distribution, we decided to extend the analysis for the gut–liver–brain axis in milk-supplemented mice. In the controls, Iba-1^+^ microglial cells represented approximately 5% of cortico-cerebral cells, whereas this percentage was significantly enhanced to 9% after milk administration (Figure 6A–C). In the cerebellum, Iba-1^+^ microglial cells represented approximately 4% of total cerebellar cells in the controls, while in milk-supplemented mice, this percentage increased to 10% of cerebellar cells (Figure 6D–F). The increase in microglial cells indicated that milk induced neuroinflammatory reactions in BALB/c mice.

In parallel, it was also observed that NOS2^+^ Purkinje-like cells were also significantly enhanced in the cerebellum of mice supplemented with milk. In the controls, NOS2^+^ Purkinje cells represented approximately 30% of total cerebellar cells. In contrast, this number increased to 65% in the cow’s milk-supplemented mice (Figure 6G–I). Together, these data indicate that bovine milk intake affected microglial and Purkinje cell niches in the brain of mice, suggesting amplified neuroinflammatory signals and oxidative stress.

### 3.7. Cow’s Milk Consumption Disturbed Post-Synaptic Cells in Mice

To investigate possible mechanisms between milk consumption and autistic-like behaviors, pre- and post-synaptic proteins were monitored in the cerebral cortex of both experimental groups. In the controls, Shank-3^+^ cells were widely distributed throughout the cortical regions mostly expressed by neurons and glial cells (Figure 7A). In contrast, the cow’s milk-supplemented mice showed significant disorganization of Shank-3^+^ cells in the similar area of the brain (Figure 7B). Approximately 80% of cortical cells were positive to Shank-3 in control mice, while only 15% of these cells were marked to Shank-3 in mice that received milk (Figure 7C).

Other synaptic proteins were also evaluated in these mice. Cortical Synaptophysin^+^ and Drebrin^+^ cells were widely dispersed throughout the brain parenchyma, frequently co-localized in the tissue of control mice (Figure 7D). On the other hand, mice supplemented with milk presented a significant reduction in Drebrin immunoreactivity, with drastic consequences to co-localization of both synaptic proteins (Figure 7E). These data pointed to Shank-3 and Drebrin, both post-synaptic proteins, as potential targets to be studied in experimental models of milk supplementation and atypical behaviors in mice.

## 4. Discussion

For the first time, it was demonstrated that cow’s milk intake amplified atypical behaviors in genetically susceptible BALB/c mice, including social interactions, stereotyped movements, and restricted interest. These behavioral patterns can be associated with gut–liver–brain disorders, since these milk-supplemented mice showed significant gut–liver inflammation and disturbances on post-synaptic cell niches (Shank-3^+^ and Drebrin^+^ cells) in the cerebral cortex. Moreover, mice drinking milk showed important signals of neuroinflammation and oxidative stress in the brain. Thus, it is plausible to suggest that bovine milk intake affected the gut–brain axis in BALB/c mice.

Although we did not identify specific milk components in modulating the gut–brain axis, it was clear that BALB/c mice supplemented with milk showed important behavioral, cellular, and molecular changes. The experimental condition, based on UHT milk administration, can be correlated with humans, because both mice and humans have similar histological architecture in the gut, they exhibit lactase in their intestinal microvilli, and conservative transport machinery to transfer sugars from the lumen to the bloodstream. On the other hand, they differ in microbiome and biochemical mechanisms linked to digestive functions [18,19]. Research involving cow’s milk and intestinal disorders in BALB/c mice covers various aspects, including lactose intolerance, allergic reactions, gut barrier function, and microbiota alterations [20,21,22]. In this work, we did not identify specific milk components in modulating the gut–brain axis, but it was very clear that BALB/c mice supplemented with milk showed important behavioral, cellular, and molecular changes.

The gut–brain axis refers to the bidirectional communication network linking the gastrointestinal tract and the brain. Milk contains several components that can influence this axis, such as proteins (like casein) [23], fatty acids (including omega-3 and omega-6) [24], lactose (sugar can affect gut microbiota composition), vitamins and minerals (like B vitamins, and calcium and magnesium), and probiotics (can positively affect gut health) [25]. Recently, Robinson and colleagues revised different mechanisms in which both A1 β-casein and A2 β-casein in the bovine milk affect the gut–brain axis [26]. Indeed, all these milk compounds can influence mood and cognitive functions through the gut–brain axis. These studies help in understanding the potential mechanisms through which cow’s milk might affect gut health and contribute to disorders.

Using BALB/c mice that naturally have autistic-like behaviors, including low sociability and high stereotypic index [11,12], we investigated whether milk potentially expands the previous behavioral patterns linked to autism and possible pathways associated with gut–brain axis disorders. Here, it was described that milk enhanced social interaction deficits (low sociability), repetitive (stereotypic) movements, and restricted interest in BALB/c mice. Our findings support a possible correlation between bioactive compounds in bovine milk, such as casein, lactose, and other glycoproteins, and autistic-like behavior [27,28]. In accordance, autistic humans are frequently submitted to a milk-free diet and there are reports of improvements in intestinal and social functions in these individuals. However, this strategy is not conclusive and there is no evidence to prescribe a cow’s milk-free diet to treat individuals with autism [29,30,31].

The possibility of cellular activation in mice supplemented with milk led us to monitor the phenotype of blood leukocytes by flow cytometry. Milk induced a significant increase in circulating granulocytes and TCD4^+^ cells. In agreement, dietary factors were able to mobilize granulocytes, including neutrophils and eosinophils, in the bloodstream after intake of proteins (e.g., casein) and carbohydrates (e.g., lactose) present in the cow’s milk [32,33,34]. In the same analysis by flow cytometry, it was observed that B lymphocytes were significantly reduced in the blood of milk-supplemented mice. Recent reviews reinforced the crucial role of B lymphocytes differentiating in plasma cells on the biology of the gut–liver axis, especially in the intestinal mucosa, where these cells secrete antibodies and regulate intestinal immune responses. Furthermore, milk-derived nutrients or bioactive compounds might influence immune signaling pathways, affecting B cell activation and differentiation [35,36]. In our experimental model, the reduction in circulating B cells in the bloodstream and the increased concentration of CD138^+^ plasma cells in the intestinal lamina propria could be linked to milk consumption.

In the gut and liver samples, we observed signals of inflammation after introducing the milk supplementation. In the small intestine, intraepithelial lymphocytes, a type of immune cell found in the epithelial layer of the gut mucosa, were substantially increased in mice supplemented with milk. These cells play a crucial role in maintaining gut homeostasis, responding to pathogens, and regulating immune responses, particularly in the context of allergies and food intolerances. The increased percentage of intraepithelial lymphocytes indicates an immune response to inflammation or injury [37]. Although our data are inconclusive in this context, our data indicated that bovine milk induced an inflammatory response in the gut of BALB/c mice. The number and distribution of KI67^+^ cells and CD138^+^ cells reinforced this proposal.

The high number of intestinal KI67^+^-proliferating cells suggests that growth factors and bioactive compounds in the milk can induce cellular proliferation [38,39]. Research with KI67^+^ cells helps to explain a paradoxical perspective indicating that milk has beneficial effects on gut repair, or it has potential risks [40,41]. In parallel, we observed that CD138-expressing cells in the lamina propria of the small intestine were significantly disturbed in milk-supplemented mice. CD138 (syndecan-1) expression can be enhanced in response to intestinal inflammation, particularly when IgA-secreting plasma cells are induced to protect the intestinal mucosa from pathogens and maintain mucosal immunity [42].

It is critical to note that UHT milk undergoes a process that heats the milk to at least 135 °C (275 °F) for a few seconds to kill harmful bacteria and extend shelf life. However, this process can impact the bioactivity of certain components in the milk [43]. During UHT processing, milk can lose nutrient stability, such as reducing levels of some vitamins, but most essential nutrients like calcium and protein can remain stable. Lactose content also remains largely unaffected, making UHT milk suitable for those who can tolerate lactose [44]. Overall, while UHT milk seems safe and convenient, some of its bioactive properties may be compromised compared to raw or pasteurized milk. These data reinforced that our experimental model is plausible to investigate gut–brain axis functions in mouse models [45].

In the liver, milk supplementation was associated with oxidative, inflammatory, and atypical metabolic signals. The high expression of hepatic nitric oxide synthase 2 (NOS2), also known as inducible nitric oxide synthase (iNOS), can be linked to acute inflammatory response triggered by pro-inflammatory cytokines, resulting in oxidative stress, apoptosis, and fibrosis [46,47,48]. The pro-inflammatory conditions also impact glycogen metabolism, leading to abnormal glycogen synthesis and storage by hepatic cells [49], and gluconeogenesis to supply energy during inflammation [50]. Consistently, our data indicated that atypical accumulation of both NOS2 and glycogen in the liver of BALB/c mice supplemented with cow’s milk should be critical tissue markers to monitor possible liver damages in this experimental model.

To investigate possible molecular mechanisms affected in the gut–liver axis after milk consumption, the RNAm expression of pro-inflammatory genes were evaluated by RT-PCR. Increased levels of IL-1β, a key pro-inflammatory cytokine modulated by dietary components [51], has been linked to hepatocyte injury and fibrogenesis [52]. Paradoxically, IL-1β can be inhibited by bioactive milk proteins [53] or to induce pro-inflammatory signals [54]. In our experimental conditions, the IL-1β expression was significantly increased in the liver of milk-supplemented mice, suggesting that milk intake favored pro-inflammatory signals in the gut–liver axis.

Other target genes were also overexpressed in mice supplemented with cow’s milk. In our experimental model, the mRNA expression of CCL2 (C-C motif chemokine ligand 2), CXCL1 (C-X-C motif chemokine ligand 1), CYP1A2 (cytochrome P450 1A2), and GRα (Glucocorticoid receptor alpha) were also significantly increased in the liver after milk consumption. CCL2, also known as MCP-1 (monocyte chemoattractant protein-1), is often upregulated during chronic liver diseases [55] and linked to intestinal epithelial damages [56]. CXCL1 expression is also elevated in acute injury and chronic liver diseases, frequently associated with fibrosis and cirrhosis [57]. During gut barrier disorders, liver sinusoidal endothelial cells can assume hepatic barrier functions, but they increase the expression of CXCL1 in the liver [58]. In our model, the enhanced expression of these genes indicated an inflammatory reaction after milk intake.

CYP1A2 plays a crucial role in the metabolism and detoxification of various substances in the liver, including drugs, environmental toxins, and endogenous compounds [59]. Its overexpression was also related with gut–liver axis perturbations [60]. Finally, the hepatic GRα expression is associated with anti-inflammatory pathways, by inhibiting the expression of pro-inflammatory cytokines and chemokines [61]. In contrast, elevated glucocorticoid levels can induce changes in the gut permeability [62]. Together, our data suggest that the gut–liver axis in BALB/c mice was significantly affected by milk consumption and systemic effects should be monitored in these animals.

Significant signals of neuroinflammation and oxidative stress in the CNS were also observed in these milk-supplemented BALB/c mice. The increase in Iba-1^+^ cells in the cerebral and cerebellar cortex suggested that microglial cells can be activated by peripheral inflammatory stimuli [63]. Consistently, the activation and/or increase in Iba-1^+^ microglial cells are frequently observed during neuroinflammatory responses, and it can be associated with cognitive impairment [64]. Furthermore, cerebellar oxidative stress has been implicated with impaired functions, given that cerebellum plays a crucial role in motor control and cognitive functions [65]. Oxidative stress can lead to neuroinflammation, and in the cerebellum, this can affect neuronal health and function [66]. These events have also been associated with atypical behaviors, including those observed in autism [67,68]. Understanding these mechanisms is critical for developing targeted interventions and therapies.

Among other experimental models, BTBR T + tf/J (BTBR) mice have attracted attention for similarities with ASD in humans [69]. Interestingly, these BTBR mice were also characterized by elevated numbers of Iba-1^+^ cells and oxidative stress in the cerebellum, both correlated with stereotypies [70]. Our data corroborated with these findings. The cerebellum of milk-supplemented mice was also characterized by increased numbers of Iba-1^+^ microglial cells and NOS2^+^ Purkinje cells. In addition, the increased numbers of cerebellar NOS2^+^ Purkinje cells in milk-supplemented mice suggested that cow’s milk can induce oxidative stress in this area of the brain. These data incite a hypothesis that early damage outside of the CNS can be a trigger to initiate behavioral changes.

Although it is very difficult to correlate behavioral changes with morphological and biochemical data, the distribution of synaptic proteins reveals some evidence. In this context, Shank-3, Drebrin (both post-synaptic proteins), and Synaptophysin (pre-synaptic protein) were monitored in our experimental conditions. Our results indicated that consumption of milk affected two important post-synaptic proteins, Drebrin and Shank-3. Shank-3 is a crucial scaffolding protein located at the post-synaptic density of neurons, involved in synaptic signaling and plasticity [71]. Mutations or deletions in the *SHANK3* gene are linked to neurodevelopmental disorders, including ASD and striatal dysfunctions [72]. These Shank-3^−/−^ mice exhibit several behaviors that are reminiscent of ASD symptoms, such as social interaction deficits, repetitive behaviors, anxiety-like behaviors, and communication deficits [72,73].

Drebrin is involved in the structural dynamics of dendritic spines, which are critical for synaptic connections and plasticity. It helps regulate actin filaments in the post-synaptic density, influencing the stability and function of synapses [74]. Its relevance to ASD has been increasingly recognized in research. Studies have shown that drebrin levels may be altered in individuals with autism, potentially impacting synaptic development and function [75]. Understanding the roles of Shank-3 and Drebrin in synaptic function helps elucidate the biological underpinnings of autism and may establish the way for novel therapeutic approaches.

These results were detailed to monitor the milk pathways and possible interferences with the gut–brain axis. Given that BALB/c mice are genetically predisposed to develop patterns of behaviors compatible with autism, we proposed that UHT milk potentialized some behavioral symptoms in these mice. In parallel, histological and physiological disorders were observed in the gut, liver, and brain of milk-supplemented mice. Although mechanisms continue to be poorly understood, the modulation of the expression of pro-inflammatory signals, oxidative stress, and neuroinflammation co-localized with reduced expression of Shank-3 and Drebrin seems substantially promising.

## 5. Conclusions

Our data indicated that mice receiving cow’s milk by oral gavage spent more time performing stereotypes, social retraction with non-familiar mice, and contacting a familiar mouse in three chambers test. On the other hand, these mice supplemented with milk showed even less sociability, including low affinity by same-lineage females, and less time exploring new objects than controls, suggesting restricted interest. Importantly, milk consumption did not interfere with motor parameters. These behavioral parameters were associated with significant histological and physiologic disturbances, such as a reduction in cells expressing synaptic proteins after cow’s milk intake. Our data pointed to important cell targets in the gut–brain axis potentially linked to atypical behavioral patterns.

## Figures and Tables

**Figure 1 biomedicines-12-02448-f001:**
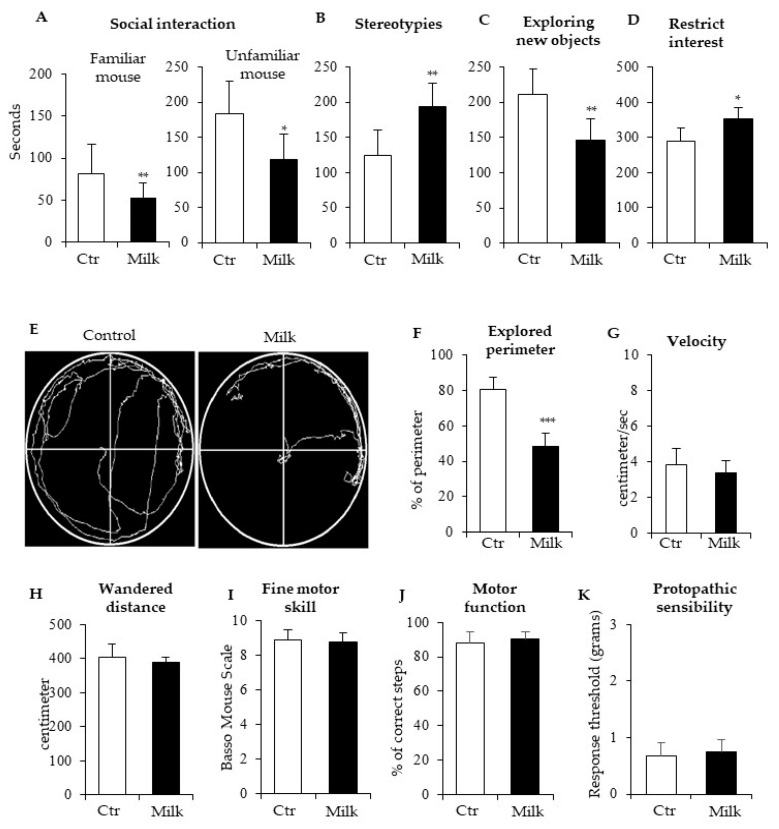
Behavioral pattern of BALB/c mice supplemented with cow’s milk. (**A**–**D**) Autistic-like behaviors were analyzed in BALB/c mice and the time spent in social interaction with a familiar or unfamiliar mouse was evaluated (**A**): (**B**) stereotyped repetitive movements, (**C**) time exploring new objects, and (**D**) restricted interest were also measured in control (ctr) and milk-supplemented mice (milk). The perimeter of the circular arena was monitored for both mice (**E**). The global motility test was used to investigate total perimeter (**F**), velocity (**G**), and distance (**H**) during the travel through the arena. These important tests also revealed that milk consumption did not influence fine motor skills (**I**), motor functions (**J**), or protopathic sensibility (**K**). White bars represent control mice (supplemented with water) and black bars indicate milk-treated mice values. These data are representative of three independent experiments. (*) *p* < 0.05; (**) *p* < 0.01, (***) *p* < 0.001.

**Figure 2 biomedicines-12-02448-f002:**
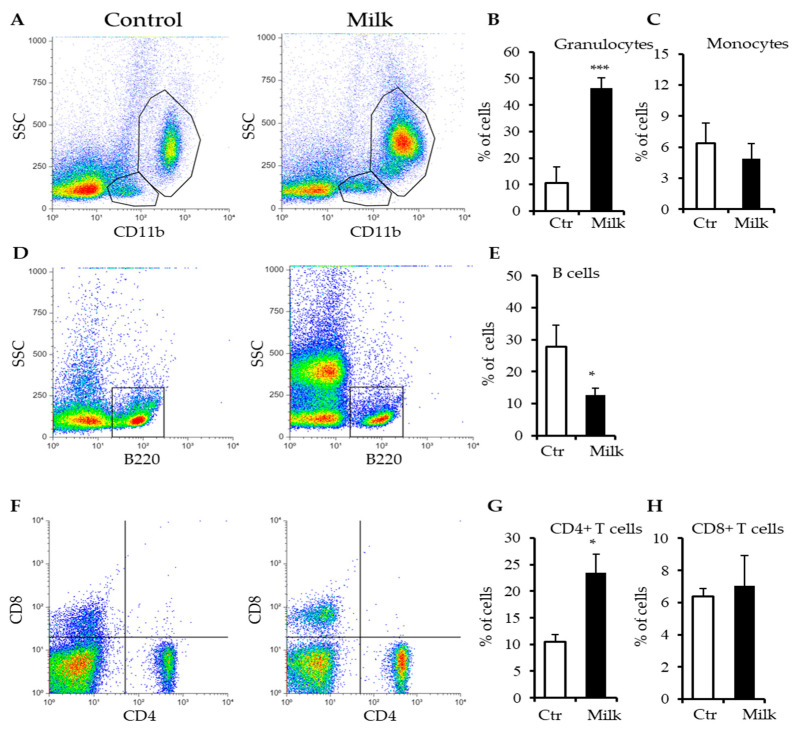
Phenotypical and quantitative analysis of blood leukocytes of mice supplemented with cow’s milk. By flow cytometry, blood leukocytes were analyzed with phenotyping as follows: (**A**) Myeloid cells were gated by the expression of CD11b. These cells were subdivided into CD11b^+^SSC^high^ granulocytes (up gate) and CD11b^+^SSC^low^ monocytes (down gate). They were quantified in (**B**,**C**), respectively. (**D**) B lymphoid cells expressing B220 (B220^+^SSC^low^) were selected (cubic gate) and quantified in (**E**). (**F**) T lymphoid cells expressing CD4 or CD8 were subdivided into CD4^+^CD8^−^ T helper cells and CD4^−^CD8^+^ T cytotoxic cells. They were quantified in (**G**,**H**), respectively. Control indicates mice receiving water (white bars) and Milk indicates mice supplemented with cow’s milk (black bars). Data are representative of three independent experiments. (*) *p* < 0.05; (***) *p* < 0.001.

**Figure 3 biomedicines-12-02448-f003:**
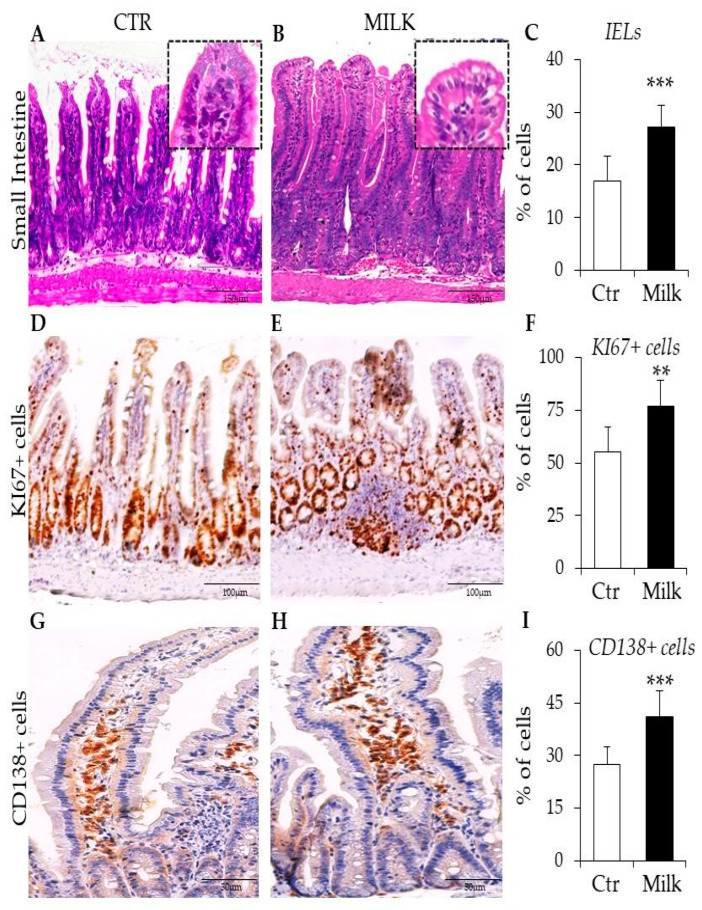
Morphological analysis of the small intestine in mice supplemented with cow’s milk. Representative photomicrographs of the small intestine obtained from mice receiving water ((**A**): CTR—controls) or milk ((**B**): MILK). Inserts show detailed images of the villi. (**C**) Quantitative analysis of intraepithelial lymphocytes (IELs) in the intestinal epithelial tissue of the mucosa. Immunohistochemistry staining to KI-67 localizes proliferative cells in the gut mucosa in control (**D**) and milk-treated mice (**E**). (**F**) Quantitative analysis of KI-67 expressing cells in the mucosal epithelial cells. Immunohistochemistry staining to CD138 localizes immunoglobulin secreting plasma cells in villi of control (**G**) and milk-treated mice (**H**). (**I**) Quantitative analysis of CD138-expressing cells in the connective tissue of the mucosa. Data are representative of three independent experiments. Amplification: 1000× (inserts). (**) *p* < 0.01; (***) *p* < 0.001.

**Figure 4 biomedicines-12-02448-f004:**
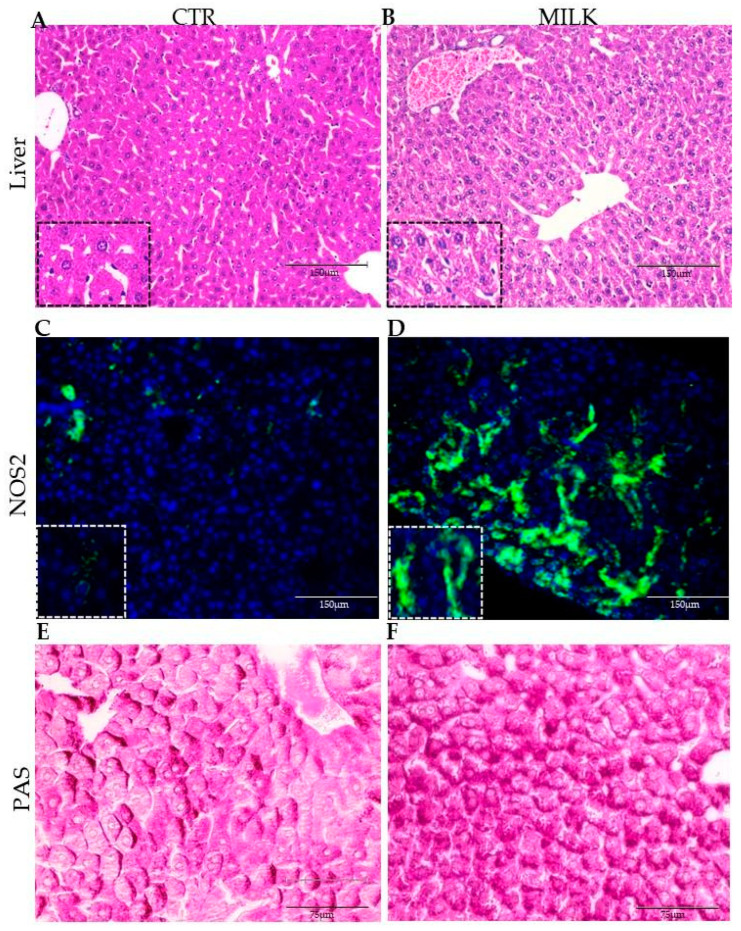
Morphological analysis of the liver in mice supplemented with cow’s milk. Representative photomicrographs of the liver obtained from mice receiving water ((**A**): CTR—controls) or milk ((**B**): MILK). Inserts (**A**,**B**) show detailed images of the lobular zone. Immunofluorescence analysis staining to NOS2 enzyme localizes oxidative stress niches in the liver of control (**C**) and milk-treated mice (**D**). Inserts (**C**,**D**) show detailed images of hepatic cells in the oxidative stress niches. PAS staining identifies glycogen accumulation by hepatic cells in control (**E**) and milk-treated mice (**F**). Data are representative of three independent experiments. Amplification: 100× (**A**–**D**); 400× (**E**,**F**); 1000× (inserts).

**Figure 5 biomedicines-12-02448-f005:**
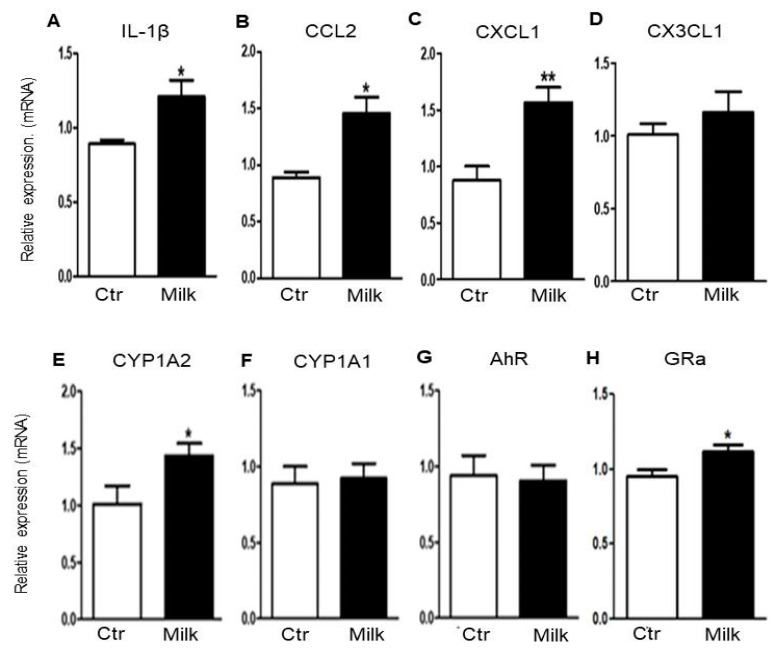
Quantitative analysis of mRNA gene expression by hepatic cells of mice supplemented with cow’s milk. Bar graphs represent the mRNA gene expression by RT-PCR on hepatic cells of mice receiving water (CTR—controls, white bars) or milk (MILK, black bars). (**A**) Interleukin 1-beta (IL-1β); (**B**) C-C motif ligand 2 (CCL2); (**C**) Chemokine (C-X-C motif) ligand 1 (CXCL1); (**D**) C-X3-C motif chemokine ligand 1 or fractalkine (CX3CL1); (**E**) Cytochrome P450 1A2 (CYP1A2); (**F**) Cytochrome P450 family 1 subfamily A member 1 (CYP1A1); (**G**) Aryl hydrocarbon receptor (AhR); (**H**) Glucocorticoid receptor agonist (GRa). Data are representative of three independent experiments. (*) *p* < 0.05; (**) *p* < 0.01.

**Figure 6 biomedicines-12-02448-f006:**
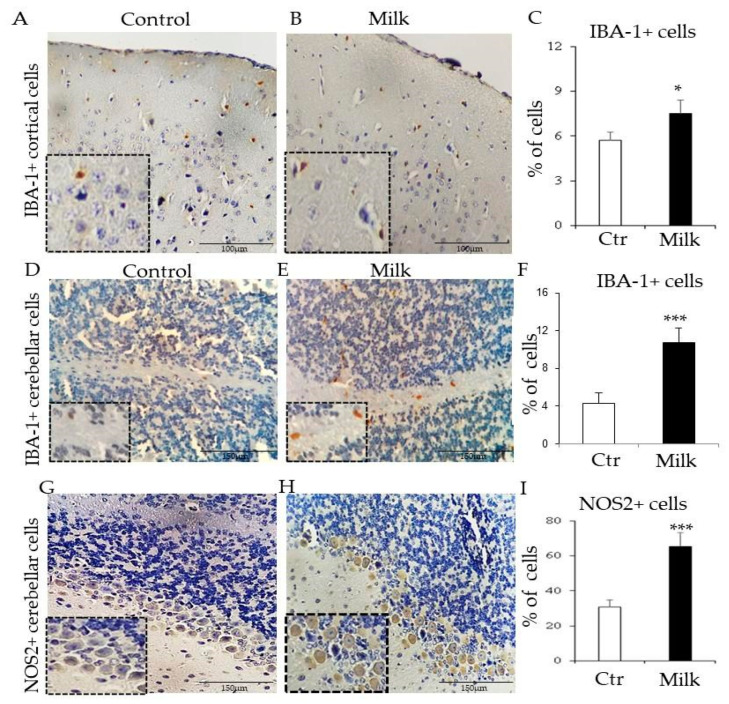
Localization of Iba-1^+^ and NOS-2^+^ cells to link neuroinflammation and oxidative stress with cow’s milk intake in mice. Immunohistochemistry staining to Iba-1 localizes cortical microglial cells in control (**A**) and milk-treated mice (**B**). The bar graphs indicate the percentage of cortical cells expressing Iba-1 (**C**). Cerebellar Iba-1^+^ microglial cells were localized in controls (**D**) and milk-treated mice (**E**). The bar graphs indicate the percentage of cerebellar cells expressing Iba-1 (**F**). NOS-2^+^ cells related to oxidative stress were localized in controls (**G**) and milk-treated mice (**H**). These NOS-2^+^ cells showed morphology compatible with Purkinje cells ((**G**,**H**), inserts) and they were quantified in the cerebellum (**I**). White bars: control group (Ctr). Black bars: milk-treated mice. Black boxes highlight amplified images of Purkinje cells. These data are representative of three independent experiments. (*) *p* < 0.05; (***) *p* < 0.001.

**Figure 7 biomedicines-12-02448-f007:**
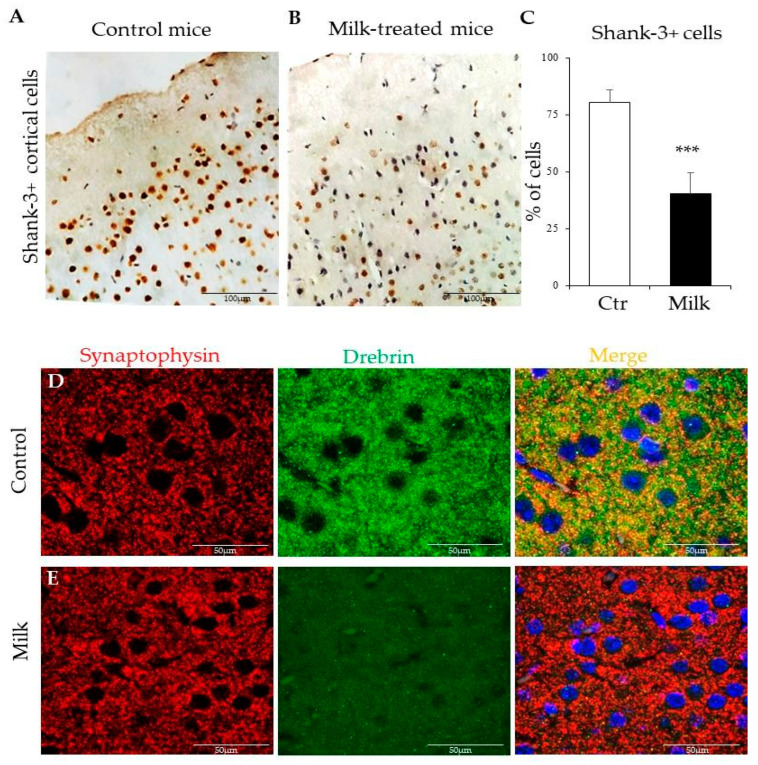
Immunohistochemical and immunofluorescence analyses of cerebral cortex in BALB/c mice supplemented with cow’s milk. Immunohistochemistry staining localizes Shank-3^+^ cells in the cerebral cortex of control (**A**) and milk-treated mice (**B**). The bar graphs indicate the percentage of cortical cells expressing Shank-3. White bars represent mice supplemented with water (controls) and black bars indicate milk-treated mice values (**C**). Immunofluorescence microscopy revealed Synaptophysin^+^ cells (red) and Drebrin^+^ cells (green) in the cortex of control (**D**) and experimental group (**E**). In both, merge represents an overlay of images (preferentially orange in (**D**) and red in (**E**)); in blue, nucleus. These data are representative of three independent experiments. Amplification: 200× (**A**,**B**); 400× (**D**,**E**). (***) *p* < 0.001.

## Data Availability

The data used to support the findings of this study are available from the corresponding author upon request.

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
