# Peer review of "UHT Cow’s Milk Supplementation Affects Cell Niches and Functions of the Gut–Brain Axis in BALB/c Mice"

_biomedicines, 2024, doi:10.3390/biomedicines12112448_

Round 1

Reviewer 1 Report

Comments and Suggestions for Authors

This study investigates the impact of UHT cow's milk supplementation on the gut-brain axis and behavioral patterns in BALB/c mice, revealing that milk consumption disrupts cellular niches, induces neuroinflammation and oxidative stress, and leads to atypical behaviors associated with autism-like signals. Here are some comments.

  1. The manuscript presents interesting findings on the effects of UHT cow's milk on the gut-brain axis in BALB/c mice. However, the organization of the results section could be improved for better readability. It would be beneficial if the authors could present the data in a more logical flow, perhaps starting with the general effects and then moving into specific cellular and molecular details.
  2. Figure 1, 2, 5 and 7 need to indicate the scale.
  3. The discussion of the results is thorough; however, the authors might consider discussing potential mechanisms in greater detail. For instance, the role of specific milk components in modulating the gut-brain axis and how these might differ from other dairy products.
  4. Is it possible that the effects of milk on the intestines and brains of mice are caused by the mice's allergy to milk or other intestinal discomfort
  5. Does it make sense to choose mice to study the effects of milk in a similar way to human health

Author Response

Responses to REVIEWER #1

  1. The manuscript presents interesting findings on the effects of UHT cow's milk on the gut-brain axis in BALB/c mice. However, the organization of the results section could be improved for better readability. It would be beneficial if the authors could present the data in a more logical flow, perhaps starting with the general effects and then moving into specific cellular and molecular details.

Many thanks for this suggestion. The original sequence of figures was changed following the recommendation. Now, data explored general effects (behavioral pattern and leukocyte mobilization in the blood) and, subsequently, we demonstrated particular effects in the gut, liver and brain, respectively. The new format seems more interesting.

Figure 1 – old Figure 6

Figure 2 – old Figure 4

Figure 3 – old Figure 1

Figure 4 – old Figure 2

Figure 5 – old Figure 3

Figure 6 – old Figure 5

Figure 7 – maintained as Figure 7

  1. Figure 1, 2, 5 and 7 need to indicate the scale.

I am sorry for this primary mistake. Of course, it was very important to demonstrate the scales in the photomicrographs. The scales were inserted adequately in each figure. Now, Fig 1 changed to Fig 3, Fig 2 changed to Fig 4, Fig 5 changed to Fig 6, and Fig 7 was maintained as Fig 7. Many thanks!

  1. The discussion of the results is thorough; however, the authors might consider discussing potential mechanisms in greater detail. For instance, the role of specific milk components in modulating the gut-brain axis and how these might differ from other dairy products.

Based on the suggestive sentence “the role of specific milk components in modulating the gut-brain axis and how these might differ from other dairy products”, we added a new paragraph in the discussion (Second paragraph of the discussion; page 14). Clearly, this suggestion improved our perspectives. Moreover, this aspect inserted in the discussion can be studied in the future by our team. Thanks again!

  1. Is it possible that the effects of milk on the intestines and brains of mice are caused by the mice's allergy to milk or other intestinal discomfort?

Excellent question! First, I would like to mention that our protocol differs from those used to generate food-allergy models. Moreover, we didn't observe the expansion of eosinophils in intestinal mucosa. Our data are insufficient to propose allergy and intestinal discomfort. During behavioral tests, these mice did not show signal os pain. We added a new result in the figure 3 (currently, regarding the small intestine) after reanalysis of the images of the intestinal mucosa. Mice supplemented with milk had signals of inflammatory reactions, such as increased number of intraepithelial lymphocytes (new data), KI67+, and CD138+ cells. Topic 3.2 of the results was substantially changed to include these data and some references to support these tests. In accordance, we discussed these findings better in the last paragraph, page 16, discussion section.

  1. Does it make sense to choose mice to study the effects of milk in a similar way to human health?

New sentences were inserted in the 2nd, 3rd and 4th paragraphs of the discussion section to improve our choice . It was very important to question the mouse model. Mice, just like humans, exhibit lactase in their microvillus, as well as conservative transport machinery to transfer sugars from the lumen to the bloodstream. Moreover, BALB/c mice are considered neuroatypical models because they naturally show autistic-like behaviors. In parallel, humans diagnosed with autism are frequently submitted to a milk-free diet and part of them improve their behavioral symptoms. In our experimental conditions, we decided to investigate the potential of milk as a trigger to expand the previous behavioral patterns linked to autism and to investigate possible pathways in the gut-brain axis

Reviewer 2 Report

Comments and Suggestions for Authors

Could you provide more details on the rationale behind using BALB/c mice specifically for this study? How do their baseline behavioral traits compare to other mouse strains, and why were they chosen for investigating cow’s milk effects on the gut-brain axis?

How might the processing of cow’s milk at ultra-high temperatures (UHT) influence its bioactivity compared to non-UHT processed milk? Could this processing method have contributed to the observed inflammatory and oxidative stress responses?

The control group received only water. Would it have been beneficial to include a group receiving a non-dairy milk alternative to differentiate between the effects of cow’s milk specifically and general liquid consumption?

 Can you elaborate on how the behavioral features, such as low sociability and stereotypies, were quantitatively assessed in these mice? Were any standardized behavioral tests used to ensure objective measurement?

 The study suggests a link between cow’s milk consumption and synaptic dysfunction in the brain. Could you expand on the potential molecular mechanisms by which milk might directly or indirectly affect the synaptic niches in the brain, particularly through gut-brain axis signaling?

Given the significant behavioral and physiological effects observed in mice, how do you envision translating these findings to human health, particularly in the context of early childhood consumption of cow’s milk?

Author Response

Responses to REVIEWER #2

  1. Could you provide more details on the rationale behind using BALB/c mice specifically for this study? How do their baseline behavioral traits compare to other mouse strains, and why were they chosen for investigating cow’s milk effects on the gut-brain axis?

Thank you very much for the excellent question. New sentences were included in the 4th paragraph of the discussion. BALB/c mice were described in 2007 as neuroatypical mice (Brodkin ES. BALB/c mice: low sociability and other phenotypes that may be relevant to autism. Behav Brain Res. 2007 Jan 10;176(1):53-65. doi: 10.1016/j.bbr.2006.06.025). BALB/c was indicated as an experimental model for autism because these mice have low sociability, stereotyped movements and restricted interest. Other authors used BALB/c mice to investigate atypical behavioral patterns in the following years (Moy et al., 2008; Gandhi et al., 2021; Gouda et al., 2024). In our model, the rationale was established based on this information. Therefore, we used BALB/c mice as they already show signs of autistic-like behavior and supplemented them with cow's milk, as a possible trigger to expand symptoms. We do not understand this model as the cause of autism. We improved this information in the manuscript, following your important question

  1. How might the processing of cow’s milk at ultra-high temperatures (UHT) influence its bioactivity compared to non-UHT processed milk? Could this processing method have contributed to the observed inflammatory and oxidative stress responses?

Again, thanks for other important questions! We searched the literature for production mechanisms, as well as the processing of cow's milk at ultra-high temperatures (UHT) to link with bioactivity. We need to test different types of processed milk and to investigate other parameters, including centrifugation of UHT milk before the consumption. However, we believe that these methods can be analyzed in the future. Indeed, your suggestion led us to an internal discussion and our team is willing to carry out a comparative study with different ways of processing cow's milk. We inserted a new paragraph (3rd paragraph, page 17, discussion).

  1. The control group received only water. Would it have been beneficial to include a group receiving a non-dairy milk alternative to differentiate between the effects of cow’s milk specifically and general liquid consumption?

The definition of the control group was widely discussed within our team during the experimental design phase. The decision to use water was intended to offer the same factors that could temporarily stress the animals (holding for gavage, for example), maintaining the same volume of supplemented liquid. Furthermore, we chose a liquid that would not generate an inflammatory response in the digestive tract. We thought about other types of nutrients, but we were stuck on the possibility of creating other variables due to the composition of other potential controls. Thus, the control group received only water to avoid the risk of presenting systemic signs resulting from other factors. Without a doubt, the idea is excellent and, as stated in the previous answer, we can include a new methodology to evaluate alternative diets to cow's milk in our team's next steps in this line of scientific research, indeed, to complement the anterior suggestion.

  1. Can you elaborate on how the behavioral features, such as low sociability and stereotypies, were quantitatively assessed in these mice? Were any standardized behavioral tests used to ensure objective measurement?

The movements of each mouse were recorded for 600 seconds (10 minutes), after the period of adaptation to the analysis platforms. In the three-chamber test, the sociability was measured by the time spent in the chamber containing another animal (familiar or unfamiliar), or even an object previously presented to the animals. Regarding stereotypes, we used a stopwatch to add up the time spent by each animal performing a repetitive movement on the open field platform. The graphs were created, and the evaluated time was described on the "Y" axis in seconds. Data were plotted as the average of the tested populations. It is worth mentioning that the behavioral tests carried out are the same as those used by other research groups. Furthermore, we used neurotypical mice (C57BL/6) as internal controls in the experimentation room, to standardize the procedures and evaluate whether the environment was favorable for carrying out the tests. These facts were included in the methodology (item 2.2 Behavioral assays).

  1. The study suggests a link between cow’s milk consumption and synaptic dysfunction in the brain. Could you expand on the potential molecular mechanisms by which milk might directly or indirectly affect the synaptic niches in the brain, particularly through gut-brain axis signaling?

Thanks for this question. We would like to have more detailed mechanistic explanations. During the interpretation of behavioral data and experimental design to continue the study, we chose to evaluate two postsynaptic protein pathways related to autistic behavior: Shank and Drebrin. The results were surprising and unprecedented. We added a paragraph to the discussion exploring these possibilities and relating them to other nutritional factors that also interfere with synapses. We emphasize that the real mechanism has not been clarified, but it provided us with fuel for future studies that are already planned by our team, in the search for a better understanding of the effects of consuming cow's milk and the signaling of the intestine-brain axis. Possibly, in our experimental model, indirect systemic mediators, and not products from the milk itself, were preponderant and responsible for the biological shutdown of the studied synapses, even if only partially. A new paragraph was inserted (last paragraphs of the discussion.

  1. Given the significant behavioral and physiological effects observed in mice, how do you envision translating these findings to human health, particularly in the context of early childhood consumption of cow’s milk?

You question an extremely relevant and delicate point. UHT milk affected the behavior and physiological niches of the gut-brain axis, including the liver, of BALB/c mice. Already thinking about the next steps, we must analyze molecular targets in the intestine itself, such as the epithelial barrier, the microbiota, and possible cellular activations in the mucosa. It is also very important to evaluate neurotypical mice, repeating the supplementation model and observing similar signaling pathways. In this way, I imagine that these discoveries can bring data from this basic research closer to human health. I hope to obtain more information, particularly in the context of cow's milk consumption, comparing different experimental models and sources of the milk itself. One possibility is to carry out molecular mapping during early childhood (in humans) to ensure that a certain type of processed milk or similar diet will not cause disturbances in the gut-brain axis. I think it's delicate, because depending on the factors, breast milk should also be tested. And in cases of suspected neurodevelopmental disorders, such as autism, a possible implementation of a diet free of milk and dairy products in early childhood, in order not to increase triggers related to behavioral disorders. Due to the delicate explanation, we chose to explain our rationale to the reviewer, but we suggest not adding such opinions in the text.

Round 2

Reviewer 1 Report

Comments and Suggestions for Authors

I have no comments now.